# The Structural Relationship between Exercise Passion, Sports Confidence, and Exercise Continuation Intention for Taekwondo Players: Moderating the Effect of the Coach’s Support

**DOI:** 10.3390/ijerph192315852

**Published:** 2022-11-28

**Authors:** Byung-Min Kim

**Affiliations:** Department of Physical Education, Sejong University, Seoul 05006, Republic of Korea; kimbm@sejong.ac.kr

**Keywords:** Taekwondo, exercise passion, sports confidence, intention to continue exercise, coach’s support

## Abstract

This study examined the relationship between “exercise passion”, sports confidence, exercise continuation intention, and the moderating effect of the coach’s support to provide basic data for Taekwondo players and instructors. A total of 428 data items were obtained using purposive sampling. Data were analyzed using frequency analysis, reliability analysis, correlation analysis, confirmatory factor analysis, structural equation model analysis, and moderating effect analysis via SPSS and AMOS version 24.0. It was found that “harmony passion” had a positive effect on all variables of sports confidence. Additionally, two variables of exercise passion had a positive effect on exercise continuation intention. Furthermore, sports confidence was identified as a variable that increased the intention to continue exercising. The coach’s support played a partial role as a moderating variable for exercise passion, sports confidence, and exercise continuation intention. It was concluded that the athlete’s passion for sports and sports confidence were important variables that increased Taekwondo athletes’ exercise continuation intention. Moreover, the active support and interest of a coach who is able to meet the athlete’s needs and exercise situation are also required.

## 1. Introduction

### 1.1. Research Needs and Objectives

Taekwondo, a world-renowned martial arts sport, is practiced by approximately 70 million people worldwide and is an official Olympic sport [1]. There are currently 211 member federations in the World Taekwondo Federation (WT), with the Vatican being officially approved as a member state in October 2021. Hence, the spread of Taekwondo and the number of people practicing it are increasing globally. Compared with past decades, the leveling of skills between countries is being progressively achieved, as witnessed by the Olympics and the world championship results.

With Korea being its host country, it needs to strive for the expansion of domestic and foreign Taekwondo club members and athletes and the continuous development of the sport. Accordingly, various studies on Taekwondo have been conducted, in which athletes’ psychological factors have been found to be important variables. The ability to control negative psychological factors, as well as physical ability, is strongly required to deliver the best performance [2]. However, previous studies have mainly analyzed the causal relationship between psychological variables. Therefore, the current literature on Taekwondo events and whether a coach’s support affects the relationships between other variables is insufficient.

Among these variables, passion, an attitude that enables one to pursue a goal with enthusiasm and strength [3], has been defined as a tendency to invest a great deal of time and energy in activities that an individual likes and considers to be important. Additionally, Philippe, Lecours, and Beaulieu-Pelletier [4] have divided exercise passion into two sub-variables, “harmony passion” and “obsessive passion”; they analyzed its effects on various factors, such as exercise performance, confidence, and performance. They reported that passion generally contributed positively to emotion. Harmony passion is defined as “the motivational force that leads an individual to be willing to participate in activities that he or she likes,” and obsessive passion is defined as “the motivational force to participate in activities because of the internal power and pressure to control him or her” [5]. Therefore, harmony passion leads to positive effects and healthy adaptation, but obsessive passion leads to coercive and negative effects.

Confidence, an important psychological variable for athletes, refers to the productive ability to understand, explain, predict, and control one‘s environment [6]. Vealey [7] first used the term “sports confidence” and defined it as the degree of personal certainty or belief in one‘s ability to successfully perform sports. Later, Weinberg and Gould [8] defined sports confidence as a firm belief in the successful performance of tasks. This sports confidence was known to induce positive emotions, provide energy, and improve the desire to perform tasks [9,10].

Those who regularly exercise are more likely to think positively and are highly psychologically satisfied [11]. In addition, continued participation in exercise can improve one’s physical ability, help one to perceive positive changes to one’s body, and help one feel a sense of psychological well-being by affecting their personality and personality development [12]. Furthermore, for athletes, exercise continuation intention refers to a continuous attachment to performing and participating in training regularly [13].

Previous studies have examined the relationship between exercise passion, sports confidence, and exercise continuation intention [5,14,15,16,17,18,19,20,21,22,23,24,25,26,27,28,29,30]. Korean studies on Taekwondo events [31,32,33,34,35,36] have also investigated the relationship between these variables and have reported a causal relationship between them.

Coaching behavior refers to the process of behavior that influences a group or individual to achieve a goal [37,38,39,40]. In Taekwondo, coaches and players spend more time together than coaches and players in other sports, forming a close relationship. Therefore, it is thought that the coach’s guidance method, consideration, and interest in the athletes can increase their intention to continue exercising. Moreover, the coach’s role is particularly important, as it is necessary for them to provide rapid feedback during the game because of the nature of the event.

In previous studies on quitting exercise, the importance of the coach’s support has been discussed over many years, garnering much interest as an academic field connected to psychology, and it is considered an important factor influencing exercise continuity [41,42]. Hence, both previous international [43,44,45,46,47,48,49,50] and Korean studies [51,52,53,54,55,56,57] have examined the relationship between the coach’s support and exercise continuation intention and have reported that coaching behavior had a positive effect on athletic continuity, as it provided players with enthusiasm and confidence.

Therefore, a causal relationship was suggested between exercise passion, sports confidence, and exercise continuity, as perceived by athletes. However, although various studies on this topic have been conducted, the results have mainly been derived via regression analysis. Hence, very few studies have used the coach’s support, which could have a significant effect on performance, as a moderator variable. Furthermore, although research has been conducted into various sports, comparatively few have focused on Taekwondo.

Consequently, the purpose of this study was to provide a way to increase exercise continuation intention among Taekwondo players through a structural equation model analysis between exercise passion, sports confidence, exercise continuation intention, and the moderating effect of the coach’s support. In addition, this study is meaningful as it provides the basic data to improve performance and opportunities for athletes and individuals and has implications for Taekwondo teams and instructors. The results derived from this study will provide useful information in the form of basic data to improve the athletic performance of elite Taekwondo athletes in the future.

### 1.2. Research Hypotheses and Research Model

In this study, we aimed to investigate the relationship between exercise passion, sports confidence, and exercise continuation intention among Taekwondo players, and reveal the moderating effect of the coach’s support. Hence, the hypotheses are laid out in the following sections, as follows.

#### 1.2.1. The Effect of Exercise Passion on Sports Confidence

Philippe, Vallerand, and Lavigne [58] divided exercise passion into two sub-variables: harmonious passion and obsessive passion. In addition, Park and Choi [33] investigated the relationship between exercise passion and confidence in high school Taekwondo athletes and showed that they had a positive effect on sports confidence through harmonious passion, encouraging the athletes to voluntarily participate in sports. Chang and Ahn [59] also suggested that confidence is an important factor in actively participating in training and games. Therefore, in this study, the following research hypotheses were established, on the basis of a study on the relationship between exercise passion and the sports confidence of Taekwondo athletes.

**H1:** 
*Harmony passion affects sports confidence.*


**H2:** 
*Obsessive passion affects sports confidence.*


#### 1.2.2. The Effect of Exercise Passion on the Intention to Continue Exercise

Since the passion felt for exercise continuously participates in that exercise, it is an important antecedent variable that determines the intention to continue exercising [60]. This suggests that if one has a positive attitude toward a planned goal, one will be more willing to continue to move toward it; therefore, the likelihood of continuing to participate in exercise will increase [61]. Many previous studies have reported that exercise passion has a significant effect on exercise continuity intention. Therefore, in this study, the following hypotheses were established on a theoretical basis and according to the results of previous studies.

**H3:** 
*Harmony passion affects the intention to continue exercise.*


**H4:** 
*Obsessive passion affects the intention to continue exercise.*


#### 1.2.3. The Effect of Sports Confidence on the Intention to Continue Exercising

Intention to continue exercising includes obsession, continuation, and an attachment to exercising [13]. Previous studies on sports confidence and exercise continuity intention reported that the higher the physical and mental confidence, the higher the exercise continuity intention [14,16,21,28,62,63]. This suggests that participants can continue to sustain exercise, owing to the confidence gained from development and achievement through exercise. Therefore, the following research hypothesis was established, on the basis of the results of the existing research on confidence and exercise continuity.

**H5:** 
*Behavioral attitude confidence affects exercise continuation intention.*


#### 1.2.4. The Moderating Effect of the Coach’s Support on the Relationship between Exercise Passion, Sports Confidence, and the Intention to Continue Exercising

The role of coaches in sports events is very important, and many studies have also reported on the impact of coaches on athletes. Previous studies examining the relationship between coaches’ support and exercise passion [18,39,48], sports confidence [40,43,64], and intention to continue exercising [43,57,65] have shown that the coach’s support has a positive effect on other variables. However, studies analyzing the moderating effect of the coach’s support in the relationship between psychological variables are insufficient. Therefore, the following research hypotheses were established to examine whether the coach’s support has a moderating effect on the above-stated hypotheses by creating two groups: one that perceives the effect of the coach’s support to be high and another that perceives it to be low.

**H6:** 
*The coach’s support moderates the effect of harmony passion on sports confidence.*


**H7:** 
*The coach’s support moderates the effect of obsessive passion on sports confidence.*


**H8:** 
*The coach’s support moderates the effect of harmony passion on the intention to continue with exercise.*


**H9:** 
*The coach’s support moderates the effect of obsessive passion on the intention to continue with exercise.*


**H10:** 
*The coach’s support moderates the effect of sports confidence on the intention to continue with exercise.*


The research model of this study is shown in Figure 1.

## 2. Materials and Methods

### 2.1. Research Participants

Taekwondo athletes were used as the target population. Athletes who participated in the Korean National Sports Festival were selected as the sample group, and the parameters were estimated. A purposive sampling method was chosen from among the non-probability sampling methods.

Prior to this study, a preliminary survey was conducted with 50 participants. Subsequently, the reliability and validity of the instrument were verified, and the main survey was conducted after experts were consulted. Of the 500 distributed questionnaires, 446 were collected and 428 were used for actual data analyses after 18 questionnaires were excluded, due to omissions or insincere answers. The characteristics of the study participants, as used for the data analyses, are shown in Table 1.

### 2.2. Measuring Tool

The questionnaire used in this study was prepared after the researcher modified and supplemented the questions that were used in a previous study [22,27,32,57,64,65,66,67,68,69,70,71,72], according to the researcher’s intention. The questionnaire included 14 questions on exercise passion, 9 on sports confidence, 2 on the intention to continue exercising, and 6 on the coach’s support. Responses were rated on a 5-point Likert scale. In addition, 36 items, which included 5 items that measured the participants’ general characteristics, were used.

Exercise passion was developed by Vallerand et al. [22]. The scale used by Lim [66] and Min [67] was modified and supplemented to fit this study. As per a previous study, 14 items were divided into two sub-variables: harmonious passion and obsessive-compulsive passion. The Cronbach’s α value for harmony passion was 0.836 and the Cronbach’s α value for obsessive passion was 0.923, which was very high.

The concept of sports confidence was developed by Vealey et al. [68]. The tools used in studies on Taekwondo players, such as that by Park [64], were modified to suit the current study and comprised nine questions. The Cronbach’s α value of the sports confidence variable was 0.830.

The variable for the intention to continue exercising was modified and supplemented to suit a Taekwondo player target, based on the questions developed by Corbin and Lindsey [69] and used in studies such as those by Lee [71] and by Kim and Kim [70]. The reliability coefficient for the exercise duration intention variable was 0.873.

The coach’s support, set as the moderating variable, was revised to suit the Korean population by Sarason et al. [73]. The questions used in previous research, such as those used by Kim [65], were revised and supplemented to conform to Taekwondo and consisted of six questions, similar to the existing questionnaire. The Cronbach’s α value for the coach’s support variable was found to be 0.895.

### 2.3. Validation and Reliability Evaluation

Validation of the measurement items of this study was investigated via content and construct validities. Two professors with physical education majors reviewed whether the items properly addressed the concept of measurement and conducted a confirmatory factor analysis.

The results of the confirmatory factor analysis are shown in Table 2. The process established: chi-squared test (x^2^) = 612.703, degrees of freedom (df) = 164 (*p* = 0.000), rock mass rating (RMR) = 0.048, comparative fit index (CFI) = 0.904, goodness-of-fit index (GFI) = 0.861, adjusted goodness-of-fit index (AGFI) = 0.822, normed fit index (NFI) = 0.904, and root mean square error of approximation (RMSEA) = 0.080, indicating that the research model was appropriate [74,75]. As a result of this process, two items on harmony and three items on sports confidence were deleted.

To evaluate the centralized validity, construct concept reliability (CR) and average variance extracted (AVE) values were used. All CR and AVE values were above 0.7 and 0.5, respectively. Thus, concentrated validity was secured [76]. Furthermore, the reliability analysis guaranteed the reliability of the questionnaire; the Cronbach’s α-coefficient value showed a confidence level of 0.7 or higher for all the variables [77].

### 2.4. Data Analyses

Windows for SPSS, version 24.0, and AMOS, version 24.0, were used for the data analyses. Frequency analysis was performed on the general characteristics of the study participants, and confirmatory factor analysis was performed to verify the validity of the measurement tool. In addition, Cronbach’s α coefficient was checked for reliability verification, and a structural equation model analysis was performed to understand the theoretical relationships between each variable. Correlational analysis, measurement group analysis, and multiple group analysis using structural equations were performed. All significance levels were verified at *p* = 0.05.

## 3. Results

### 3.1. Correlation Analysis

The results of the correlations of the variables are shown in Table 3. The value of the correlation coefficient for all the variables did not exceed 0.85, which indicated that there was no problem of multicollinearity [78].

In addition, the smallest value (0.546) among the AVE values was larger than the square value (0.286) of the maximum correlation coefficient. Hence, discriminant validity was also not a problem [79]. Therefore, the research model that was established secured validity in verifying the relationships between each of the variables.

### 3.2. Structural Equation Model Analysis

Table 4 shows the results of the structural equation model estimates for the research model. A CFI value above 0.90 indicated an optimum fit [80,81], while in the case of RMSEA, it was evaluated as being a suitable model if the value was below 0.080 [82]. In Table 4, x^2^ = 451.282, df = 161, x^2^/df = 2.803, RMR = 0.046, CFI = 0.938, GFI = 0.900, AGFI = 0.870, NFI = 0.907, and RMSEA = 0.065, which indicated that the research model is generally acceptable [74,75].

#### 3.2.1. Relationship between Exercise Passion and Sports Confidence

The relationship between Taekwondo players’ harmony passion and sports confidence was examined to verify Hypothesis 1, which showed statistically significant results. However, obsessive passion showed significant results only in the case of physical and mental confidence. Specifically, in the relationship between harmony passion and physical and mental confidence, values of β = 0.343 and CR = 4.386 were observed.

In the relationship between obsessive passion and sports confidence, β = 0.067 and CR = 1.793; no statistically significant results were found (*p* > 0.05).

Thus, Hypothesis 1 was confirmed, while Hypothesis 2 was rejected. Therefore, it was found that harmony passion was an antecedent variable that increased sports confidence. Furthermore, obsessive passion did not affect sports confidence (*p* > 0.05).

#### 3.2.2. Relationship between Exercise Passion and the Intention to Continue Exercise

To verify Hypothesis 2, the relationship between exercise passion and exercise continuation intention was examined, wherein β = 0.551 and CR = 4.008 (*p* < 0.01). Additionally, in the relationship between obsessive passion and exercise continuation intention, values of β = 0.285 and CR = 4.256 were observed at the 99% confidence level.

Hence, it was confirmed that the two sub-variables of exercise passion increased exercise continuation intention. Therefore, Hypotheses 3 and 4 were confirmed.

#### 3.2.3. Relationship between Sports Confidence and Exercise Continuation Intention

Hypothesis 5 was based on sports confidence and exercise continuation intention. In this relationship, sports confidence increased exercise continuation intention with β = 0.351 and CR = 3.083 (*p* < 0.001), showing a significant result.

Consequently, it was found that sports confidence increased the intention to continue exercising. Therefore, Hypothesis 5 was confirmed.

#### 3.2.4. Moderating Effect of the Coach’s Support on the Relationships between Exercise Passion, Sports Confidence, and Exercise Continuation Intention

Hypotheses 6–10 aimed to verify the moderating effect of the coach’s support. Previously, group classification was conducted according to the degree of recognition of the coach’s support, through the average value of the related questions. Before the moderating effect was verified, a multi-group analysis was performed to verify the measurement identity between the two groups. Moreover, a comparison was performed between the unconstrained and measurement models to verify the identities between each group.

The analyses showed that the unconstrained models’ values were x^2^ = 824.045 (df = 328, *p* = 0.000), RMR = 0.063, CFI = 0.889, and RMSEA = 0.045. For the measurement model, these were x^2^ = 837.531 (df = 344, *p* = 0.000), RMR = 0.053, CFI = 0.889, and RMSEA = 0.060. As shown in Table 5, the identity test for the two groups via chi-squared verification revealed that there was no statistically significant difference in the x^2^ value between them (Δx^2^ = 13.486, df = 16, *p* > 0.05). The two groups were judged to be the same and their measurement identity was secured [83].

Since the identity of the two groups was secured, the moderating effect was verified using a structural equation model that equally constrained the factor-loading between the two groups, according to the degree of coach support perception.

As shown in Table 6, the analyses showed that coach support had a moderating effect on the relationship between obsessive passion and sports confidence and on that between harmony passion and sports confidence. In the relationship between harmony passion and sports confidence, β = 0.291 and CR = 2.284 (*p* < 0.05) was recorded in the group that perceived the effect of the coach’s support to be low, and β = 0.796 and CR = 3.109 (*p* > 0.05) was recorded in the group that perceived the effect of the coach’s support to be high. In the relationship between obsessive passion and sports confidence, β = −0.017, and CR = −0.298 (*p* > 0.05) in the group that perceived the effect of the coach’s support to be low, and β = 0.218 and CR = 3.109 (*p* < 0.05) in the group that perceived the effect of the coach’s support to be high.

In addition, regarding the intention to continue exercising, the variables of sports confidence had moderating effects. In the relationship between sports confidence and intention to continue exercise, β = 0.564, and C.R. = 3.526 (*p* < 0.001) in the group that perceived coach support to be low, and β = 0.148, and C.R. = 1.203 (*p* > 0.05) in the group that perceived coach support to be high.

However, there was no moderating effect in the relationship between exercise passion and exercise continuity intention. Therefore, Hypotheses 6, 7, and 10 were validated, but Hypotheses 8 and 9 were rejected.

## 4. Discussion

This study aimed to examine the relationship between Taekwondo players’ exercise passion, sports confidence, and exercise continuation intention, and furnish basic data for Taekwondo players and leaders by analyzing the moderating effect of the coach’s support in this relationship.

### 4.1. Relationship between Exercise Passion and Sports Confidence

Exercise passion served as a driving force for athletes to continue participating in training [24]. It was also found that harmony passion had a positive effect on sports confidence. Hence, it could be inferred that the greater the Taekwondo player’s harmony passion, the more positively it affected the athlete’s sports confidence, which further affected performance and winning performance. However, in this study, obsessive passion was identified as a variable that did not affect sports confidence.

Vallerand’s [25] study partially supported the results of this study as it reported that harmony passion increased positive emotions; however, obsessive enthusiasm negatively affected positive emotions. In addition, Appleton, Hall, and Hill [26] and Curran et al. [28] reported that while harmonious enthusiasm elicited positive emotions and results regarding a desire for skills mastery, obsessive enthusiasm led to negative emotional or exhaustion experiences regarding high standards of performance set by oneself or others.

Harmony passion provides a certain value to the participatory activity itself and allows for more autonomous participation in the activity [84]. In contrast, someone with obsessive passion has the characteristic of continuing to participate in the activity while experiencing negative emotions [22,28,30]. However, Amiot, Vallerand, and Blanchard [23] reported that obsessive passion had a positive effect on self-confidence, just as it played a motivating role regarding the competition that occurred on the sports field, leading players to persist with sports activities. Exercise makes athletes work toward their goals and plays a role in controlling various emotions and reducing the negative psychological states experienced [19]. Hence, it can be interpreted that there was no need to view obsessive passion in an unconditionally negative way.

Additionally, since exercise passion makes athletes strongly move toward their goals and plays a role in regulating various emotions and reducing negative psychological states [19], obsessive passion is unconditional. Therefore, this can be interpreted as suggesting that there is no need to look at exercise passion from a negative point of view. Vallerand et al. [63,85] showed a positive relationship between harmony passion and obsessive passion as they were related to exercise performance, training, competition, and exercise continuation in sports.

Moreover, athletes who willingly participate in training, even under pressure, lack the harmony and autonomy that provides value when participating in activities. However, obsessive passion, aligned with strong dependence and obsession, affects sports confidence [86]. Since the results of this study confirmed that harmony passion is an important variable that increases sports confidence, coaches should strive to introduce various training methods and maintain personal relationships to improve athletes’ harmony passion. In addition, this study revealed no statistically significant difference between obsessive passion and sports confidence, which is believed to be due to the differences in recognizing obsessive passion in each individual.

### 4.2. Relationship between Exercise Passion and the Intention to Continue Exercising

In this study, both harmony and obsessive passion had a positive effect on the intention to continue exercising. Vallerand et al. [85] reported that obsessive passion was more effective in terms of voluntary practice participation. However, many previous studies [25,26,27,59,61] reported that it negatively affected exercise continuation intention. This was believed to be due to athletes’ potential intention to continue exercising to achieve a winning performance and career path. Mageau et al. [27] and Rip et al. [29] explained that athletes had the characteristic of experiencing negative emotions in various sports activities through obsessive passion while continuing to participate in them.

Passion is generally considered to have a positive effect on exercise continuity since it involves a strong tendency to invest time and energy in activities that the individuals enjoy [22]. In addition, passion involves a strong motivation to achieve the goals set by individuals [3]. Unlike amateurs, athletes are believed to be obsessed with winning and obsessive about doing well, which affects their performance. Therefore, obsessive passion is also considered a necessary factor to increase confidence and achieve good grades. In addition, the sports environment in Korea is considered to be the result of a tendency to naturally accept compulsive enthusiasm, as it often approaches exercise as working toward the goals of professional athletes from an early age, rather than approaching it as a hobby.

Based on the results, it was confirmed that Taekwondo players voluntarily participated in exercise activities and increased their confidence and continuity through exercise performance. This suggests that compulsory participation in exercise does not require obsessive passion stemming from anxiety, such as performance pressure, and that efforts to change it into a positive psychological state are required. One should also not overlook that obsessive passion, when used properly, can improve responsible behavior and increase the willingness to exercise.

### 4.3. Relationship between Sports Confidence and the Intention to Continue Exercising

Our examination of the relationship between sports confidence and exercise continuity intention confirmed that the higher the confidence, the higher the exercise continuity intention. Many studies have consistently shown that confidence has a positive relationship with performance in sports [16,17,20]. In a study by Choi et al. [30], sports confidence was found to have a positive effect on the tendency, possibility, and reinforcement of athletic performance ability. Furthermore, confidence was reported to be an important factor in continuing sports activities, which partially supported the results of this study. Moreover, studies on Taekwondo players [35,72,87] also reported that sports confidence increased the intention to continue exercising. Sports confidence is the confidence in the ability of individuals to succeed in competitive sports [7]; those with high confidence are interested in training and can quickly overcome failures or setbacks by setting goals [88].

Previous studies reported that athlete confidence had a positive effect on performance [14,15,17,18,20]. Sports confidence determines the choice of sports when one first encounters sports activities and determines the degree of one’s endurance in the face of difficult skills and training [6]. It can be seen that the flow of the game and maintaining an appropriate level of awakening are closely related to the emotional state, which can improve performance [89]. Additionally, it was proved that sports confidence had a positive effect on the emotions, such as athletes’ satisfaction, pride, and excitement [62].

Therefore, the confidence of athletes in sports is an essential psychological factor for successful performance. Field leaders should recognize the importance of training, counseling, and management to inspire physical and mental confidence, which can affect the outcome of the game.

### 4.4. Moderating Effect of the Coach’s Support

Finally, similar to the findings of previous studies [48,57,64,65], the coach’s support was found to have a moderating effect on the relationship between a Taekwondo player’s exercise passion, sports confidence, and intention to continue exercising. In particular, harmony and obsessive passion, which are sub-factors of exercise passion, showed a moderating effect of the coach’s support in the relationship with sports confidence.

Trouilloud et al. [47], Amorose and Anderson-Butcher [41], and Reeve [50] found that providing students with the opportunity for voluntary participation and choice, and supporting their independent problem-solving, participation, and decision-making had a crucial influence on numerous psychological factors, such as learners’ basic needs and motivations in sports and physical education. In addition, Kenow and Williams’s [43] study on the relationship between coaching evaluation and the performance of athlete confidence showed that the coach’s expectation effects were not related to athlete performance. However, Solomon’s [44] study partially supported the findings of the current study.

In these results, the fact that even if the player recognizes obsessive passion, the player who recognizes the importance of the coach’s support shows high confidence is noteworthy. This is supported by various studies [46,48,49,54,55,56,57], showing that positive leadership had a positive effect on the players’ internal motivation and created a psychological state that allowed them to exercise joyfully, which was directly related to exercise continuity. In particular, players who perceive the effect of coaching support to be low can still explain the importance of a coach’s support. This has the result that obsessive enthusiasm negatively affects confidence but recognizing the importance of the coach’s support as high has a positive effect on confidence.

Subsequently, the moderating effect of the coach’s support was also found in the relationship between sports confidence and the intention to continue to exercise. Autonomous support coaching is a way of minimizing psychological pressure and the demands on athletes while providing them with opportunities to understand their moods and make continuous choices regarding exercise performance [90]. This is also defined as an individual’s belief that a leader supports self-management, such as initiating a player’s behavior, providing them with a choice, solving autonomous problems, and participating in decision-making by understanding and recognizing a player from their viewpoint [86].

González-García et al. [39] highlighted the importance of leadership by reporting that athletes’ coping strategies and affective states might significantly differ according to the coaches’ behavior profiles, with less engaged coaching profiles resulting in the worst outcomes in terms of competition. Similarly, a study on the relationship between athletes’ perception of coaches’ leadership and their affective states during competitions highlighted that the coaches’ social support is a significant predictor of controlling pre-competition negative affect, suggesting that coaches should provide athletes with instructions, social support, feedback, and autonomy [40]. The findings of the present study are consistent with these previous studies, in that coaches’ support plays a key role in the psychological experience of athletes and in their behavior.

Therefore, if the leader’s psychological and behavioral support is applied positively to the players, obsession, a negative psychological factor, not only increases the intention to continue exercising but also has a positive effect on performance. Elite athletes who specialize in sports spend a great deal of time with their leaders. Hence, their relationship is also very close. Additionally, the leader’s interest and consideration or positive feedback and guidance make the player trust in and rely on the leader. Consequently, the coach’s support factor was judged to have a positive effect on the athlete’s exercise continuation in this study.

The findings of this study confirmed that the athlete’s harmony and obsessive passion were essential variables in terms of increasing sports confidence and Taekwondo players’ intention to exercise. The coach should understand that it is important to recognize the player’s faith and trust and to try to improve their confidence and intention to continue exercising by setting goals and training methods that are most suitable for their personal characteristics and tendencies.

## 5. Conclusions

This study aimed to investigate the relationship between exercise passion, sports confidence, and exercise continuation intention for Taekwondo players, and to yield basic data for coaches and athletes by analyzing the moderating effect of the coach’s support in this relationship. Harmony passion was found to have a positive effect on sports confidence. Furthermore, both harmony and obsessive passion increased the intent to continue exercising. Physical and mental confidence also increased athletes’ intention to continue exercising. The coach’s support was found to have a moderating effect on the relationship between harmony passion and sports confidence. It also had a moderating effect on the relationship between sports confidence and the intention to continue exercising.

This study has provided basic strategies for the field of sports study through an analysis of Taekwondo players. In addition, the authors expect that various insights will be provided by follow-up studies and that research using more diverse variables will be conducted in the future. Furthermore, the preparation of research tools for the analysis of Taekwondo events is vital for the continuous growth and development of Taekwondo, a national sport in Korea.

The study had a few limitations as well. First, the study was conducted on elite athletes in Taekwondo events. Therefore, it may be difficult to generalize these results to other sports events for Taekwondo club players. Second, in this study, detailed items of sampling were not classified. However, in future studies, more specific and practical results should be derived and classified according to more detailed items, such as competition, Poomsae, and demonstrations. Third, factors other than the variables set in this study are to be expected. Hence, it is necessary to consider the characteristics of further diverse variables in subsequent studies.

## Figures and Tables

**Figure 1 ijerph-19-15852-f001:**
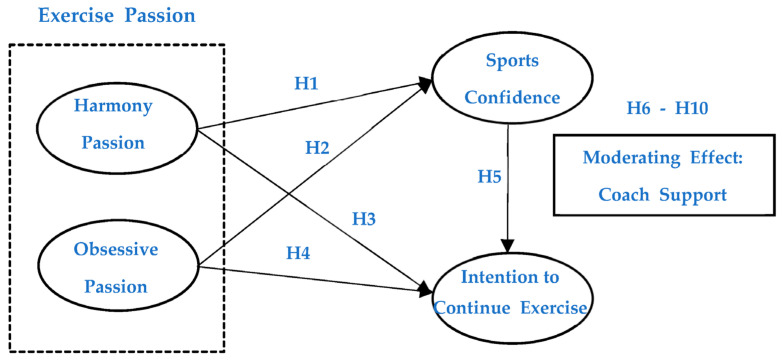
The research model of the study.

**Table 1 ijerph-19-15852-t001:** Characteristics of study participants.

Category	Content	Personnel	%
Gender	Male	277	64.7
	Female	151	35.3
Age	Less than 18	105	24.5
	18–20	104	24.3
	20–22	86	20.1
	22–24	64	15.0
	24 or more	69	16.1
Team	High school department	161	37.6
	University department	155	36.2
	General adult department	112	26.2
Athletic Career (in years)	Less than 5	58	13.5
	5–10	204	47.7
	10–15	136	31.8
	15 or more	30	7.0
Winning record	National competitions	375	87.6
	Regional competitions	34	7.9
	No winning experience	19	4.4
Total		428	100.0

**Table 2 ijerph-19-15852-t002:** Conformity verification results of the research model.

Variable	Contents	M	RW	SE	*t*	SRW	Variances	CR	AVE	Cronbach’s α
Harmony passion	2	3.80	1.000	-	-	0.616	0.333	0.887	0.611	0.836
	4	3.57	1.386	0.115	12.071	0.755	0.294			
	5	3.56	1.300	0.111	11.664	0.718	0.323			
	6	3.64	1.307	0.019	12.019	0.750	0.270			
	7	3.35	1.519	0.129	11.805	0.730	0.411			
Obsessive passion	1	2.80	1.000	-	-	0.767	0.447	0.918	0.616	0.923
	2	2.97	1.068	0.058	18.550	0.837	0.312			
	3	2.93	1.065	0.058	18.359	0.830	0.328			
	4	3.04	1.031	0.058	17.780	0.808	0.361			
	5	2.80	1.049	0.056	18.713	0.843	0.287			
	6	2.98	1.020	0.059	17.150	0.784	0.416			
	7	3.18	0.976	0.065	15.074	0.703	0.625			
Sports confidence	4	3.74	1.000	-	-	0.621	0.421	0.878	0.546	0.830
	5	3.88	0.950	0.095	10.035	0.595	0.435			
	6	3.85	1.112	0.102	10.925	0.666	0.411			
	7	3.73	1.137	0.101	11.246	0.693	0.369			
	8	3.88	1.176	0.099	11.848	0.749	0.285			
	9	3.95	1.037	0.093	11.151	0.685	0.321			
Intention to continue exercise	1	3.71	1.000	-	-	0.857	0.274	0.874	0.777	0.873
	2	3.70	1.016	0.066	15.509	0.905	0.172			
Fitness of the research model	x^2^ = 612.703, df = 164, x^2^/df = 3.736, RMR = 0.048, GFI = 0.861, AGFI = 0.822, CFI = 0.904, NFI = 0.873, RMSEA = 0.080									

Note. Total = 428; M = mean; RW = regression weights; SE = standard error; SRW = standardized regression weights; CR = critical ratio; AVE = average variance extracted; *p* = *p*-value; x^2^ = chi-squared; df = degree of freedom; RMR = root mean square residual; GFI = goodness-of-fit index; AGFI = adjusted GFI; CFI = comparative fit index; NFI = normed fit index; RMSEA = root mean square error approximation.

**Table 3 ijerph-19-15852-t003:** Results of the correlation analysis.

Variable	M	SE	1	2	3	4
Harmony passion	3.5831	0.64406	1.00			
Obsessive passion	2.9559	0.85081	0.535 **	1.00		
Sports Confidence	3.8364	0.60550	0.362 **	0.297 **	1.00	
Intention to continue exercise	3.8898	0.67336	0.363 **	0.165 **	0.362 **	1.00

Note. Total = 428; M = mean; SE = standard error; *p* = *p*-value; ** *p* < 0.01.

**Table 4 ijerph-19-15852-t004:** Verification results of the research hypotheses.

H	Path	SRW (β)	SE	CR (t)	*p*	Accept
H1	Harmony passion → Sports confidence	0.343	0.078	40.386	0.000 ***	○
H2	Harmony passion → Sports confidence	0.067	0.037	10.793	0.073	
H3	Harmony passion → Intention to continue exercise	0.551	0.137	40.008	0.000 ***	○
H4	Obsessive passion → Intention to continue exercise	0.285	0.067	40.256	0.000 ***	○
H5	Sports confidence → Intention to continue exercise	0.351	0.114	30.083	0.002 *	○
Fitness of the research model		x^2^ = 451.282, df = 161, x^2^/df = 2.803, RMR = 0.046, GFI = 0.900, AGFI = 0.870, CFI = 0.938, NFI = 0.907, RMSEA = 0.065				

Note. Total = 428; H = hypotheses; SRW = standardized regression weights; SE = standard error; CR = critical ratio; AVE = average variance extracted; *p* = *p*-value; x^2^ = chi-squared; df = degree of freedom; RMR = root mean square residual; GFI = goodness-of-fit index; AGFI = adjusted GFI; CFI = comparative fit index; NFI = normed fit index; RMSEA = root mean square error approximation; ○ = hypothesis was accepted; *p* = *p*-value; *** *p* < 0.001, * *p* < 0.05.

**Table 5 ijerph-19-15852-t005:** Identification results.

Factor	x^2^	df	Δx^2^
Unconstrained model	824.045	328	Δx^2^ = 13.486
Measurement model	837.531	344	df = 17, *p* > 0.05

Note. Total = 428; x^2^ = chi-squared; df = degree of freedom; *p =* 0.637.

**Table 6 ijerph-19-15852-t006:** Verification of the moderating effect of the coach’s support.

H	Path	Low-Support Group β (High-Support Group β)	SE	CR (t)	Moderating Effect
H6	Harmony passion → Sports confidence	0.291 (0.796)	0.127 (0.150)	2.284 * (3.109)	○
H7	Harmony passion → Sports confidence	−0.017 (0.218)	0.057 (0.070)	−0.298 (3.109 *)	○
H8	Harmony passion → Intention to continue exercise	0.592 (0.667)	0.211 (0.223)	2.799 ** (2.994 *)	
H9	Obsessive passion → Intention to continue exercise	0.295 (0.291)	0.094 (0.101)	3.136 * (2.884 *)	
H10	Sports confidence → Intention to continue exercise	0.564 (0.148)	0.160 (0.123)	3.526 *** (1.203)	○

Note. Total = 428; SE = standard error; CR = critical ratio; ○ = had a moderating effect; *p* = *p*-value; *** *p* < 0.001, ** *p* < 0.01, * *p* < 0.05.

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
