# Peer review of "The Structural Relationship between Exercise Passion, Sports Confidence, and Exercise Continuation Intention for Taekwondo Players: Moderating the Effect of the Coach’s Support"

_ijerph, 2022, doi:10.3390/ijerph192315852_

Round 1
Reviewer 1 Report
Thanks for providing me the opportunity to review this manuscript.
Here below I added some suggestions that should be addressed to improve the quality of the manuscript:
- In the abstract. It is needed to describe the sample.
- In the introduction it is needed to group the information in paragraphs as well as explain the theories of each of the variables that are included in the study.
- In the introduction it is needed more rationale to justify the novelty and the plus value of this study.
- The concept of coach support needs more explanation. See here some studies that talk about it:
González-García, H., Martinent, G., & Nicolas, M. (2022). Relationships between coach’s leadership, group cohesion, affective states, sport satisfaction and goal attainment in competitive settings. International Journal of Sports Science & Coaching, 17(2), 244–253. https://doi.org/10.1177/17479541211053229
González-García, H., Martinent, G., & Nicolas, M. (2022). A Temporal Study on Coach Behavior Profiles: Relationships With Athletes Coping and Affects Within Sport Competition. Journal Of Sport &Amp; Exercise Psychology, 44(2), 94-102.
González-García, H., Martinent, G., & Nicolas, M. (2021). Relationships between perceived coach leadership and athletes’ affective states experienced during competition, Journal of Sports Sciences, 39:5, 568-575, DOI: 10.1080/02640414.2020.1835236https://doi.org/10.1123/jsep.2021-0071
- APA format should be checked in tables.
- Line 164. You referred to a previous study but you did not cite this study. please, add the quotation.
- In the section of "measuring tools" it should be added the Cronbach alpha an the psychometric characteristics of each variable.
- In the discussion section it is needed to group the information in paragraphs.
- It is needed to justify in the discussion why you think you have reached these results.
- Line 387-391. It is needed more justification in terms of coach support rationale. Please, check previous literature.

Author Response
Thank you for your kind comments. I feel that the manuscript has improved substantially in many aspects owing to your support. Please refer to each of the responses below for revision details and see the changes within the manuscript highlighted in red.
Comments and Suggestions for Authors
- In the abstract. It is needed to describe the sample.
: I have inserted the following sentences in the introduction section (page 1, line 9).
“A total of 428 data were obtained using purposive sampling“
- In the introduction it is needed to group the information in paragraphs as well as explain the theories of each of the variables that are included in the study.
: As suggested, I have grouped the information and explained the variables. (page 2-4, line 94-182)
- In the introduction it is needed more rationale to justify the novelty and the plus value of this study.
: As suggested, I have included more rationale in the introduction (page 1, line 33~41, page 2, line 63~66, line 91-93).
- The concept of coach support needs more explanation. See here some studies that talk about it.
González-García, H., Martinent, G., & Nicolas, M. (2022). Relationships between coach’s leadership, group cohesion, affective states, sport satisfaction and goal attainment in competitive settings. International Journal of Sports Science & Coaching, 17(2), 244–253. https://doi.org/10.1177/17479541211053229
González-García, H., Martinent, G., & Nicolas, M. (2022). A Temporal Study on Coach Behavior Profiles: Relationships With Athletes Coping and Affects Within Sport Competition. Journal Of Sport &Amp; Exercise Psychology, 44(2), 94-102.
González-García, H., Martinent, G., & Nicolas, M. (2021). Relationships between perceived coach leadership and athletes’ affective states experienced during competition, Journal of Sports Sciences, 39:5, 568-575, DOI: 10.1080/02640414.2020.1835236https://doi.org/10.1123/jsep.2021-0071
: Thank you for your guidance.
I referred to this and further described the contents in the introduction and discussion
I added explanations on coach support using the above references. reference [38], [39], and [40].
- APA format should be checked in tables.
: Modified to match table APA form (all tables)
Line 164. You referred to a previous study but you did not cite this study. please, add the quotation.
:Add a previous study citation to the content
- In the section of "measuring tools" it should be added the Cronbach alpha and the psychometric characteristics of each variable.
: Conbah'a for each variable was entered, and the characteristics of the variable were described separately with the hypothesis in the introduction section. (page 6, line 216-224)
- In the discussion section it is needed to group the information in paragraphs.
- The discussion was divided into four sections and the corresponding contents were described.
The revised part was marked in red
- It is needed to justify in the discussion why you think you have reached these results.
: The contents of the discussion were classified by content, and the overall contents were added after reviewing more previous studies.
- Relationship between Exercise Passion and Sports Confidence
- Relationship between Exercise Passion and Intention to continue exercise
- Relationship between Sports Confidence and Intention to continue exercise
- Moderating Effect of Coach Support
The revised part was marked in red
- Line 387-391. It is needed more justification in terms of coach support rationale. Please, check previous literature.
: In addition to the contents pointed out, previous studies were additionally cited in the introduction and discussion.
Thank you very much for your detailed review and it helped me a lot in writing my thesis.
Reviewer 2 Report
Overall, the manuscript is well done and the results are presented in a manner that is easy to understand.
There is one correction that needs to be addressed. On line 138 on Figure 1 it shows 'Exercise Passio'. A 'n' needs to be added to Passio.
Author Response
Thank you for your kind comments. I feel that the manuscript has improved substantially in many aspects owing to your support.
* Many parts were revised according to the opinions of other reviewers. Please review it positively.
Reviewer 3 Report
Thank you for you submission. Here are some comments that hopefully will improve your article. 1.The purpose of the study should have been placed earlier in the introduction. The introduction should have been longer and contained more up-to-date studies.
There are far too many hypotheses. (21 hypotheses) You have to reduce them. This will improve the readability of the article. I am not sure that all of the hypotheses are supported by the theory in the introduction.
Table 2. It is not normal to list all of the contents in each factor. Only the factor is enough.
Table 3 should also contain the mean, standard deviation and alpha value for each factor.
The hypostasized model should have been listed in the results with paths and beta values. (and significance).
Because you first have to reduce the hypotheses I don’t want to comment on the discussion and conclusion.
Good luck with your work
Author Response
Thank you for your kind comments. I feel that the manuscript has improved substantially in many aspects owing to your support. Please refer to each of the responses below for revision details and see the changes within the manuscript highlighted in red.
- The purpose of the study should have been placed earlier in the introduction. The introduction should have been longer and contained more up-to-date studies.
: Contents such as research purpose and necessity were reinforced, and additional previous studies were cited. In addition, the psychological characteristics and hypotheses of each variable were described separately.
In addition, the introduction was additionally described by classifying the research hypothesis by variable.
(page 1, line 33- 41)
- There are far too many hypotheses.(21 hypotheses)You have to reduce them. This will improve the readability of the article. I am not sure that all of the hypotheses are supported by the theory in the introduction.
: The hypotheses were reduced to 10, and after statistical analysis, research results, discussions, and conclusions were all modified. (page 3-4, line 93-148)
Accordingly, statistical processing results, research results, and discussion contents were modified a lot.
Table 2. It is not normal to list all of the contents in each factor. Only the factor is enough.
:Revision has been made as suggested. Please refer to Table 2.
Table 3 should also contain the mean, standard deviation and alpha value for each factor
: Revision has been made as suggested. Please refer to Table 3.
It was reconstructed according to the APA format.
The hypostasized model should have been listed in the results with paths and beta values. (and significance.)
: Unmarked statistical figures are described in the table and body contents. (page 8-10, line 280-358)
Because you first have to reduce the hypotheses I don’t want to comment on the discussion and conclusion.
The revised part was marked in red.
I also worked hard on the discussion in this study.
Please consider it positively.
Thank you very much for your detailed review and it helped me a lot in writing my thesis.
Round 2
Reviewer 3 Report
Thank yor for youre resubmission. You have done a lot to improve youre article, but still something is missng.
In the indtroduction you should explain the difference between harmony and obsessive passion more carefully. I think that to increase the number of Taekwondo players and club members and that Taekwondo becomes world-famous through this study is too ambitious.
Still I think there is too many hypothesis. You can remove hypothesis 3, 4 , 8 and 9.
The model dos not support these hypotheis.
Table 2 : Iyt is enough to report the mean value of each variable.
After you have reduced the hypothesis you must cahnge the discussion as well.
Good luck
Author Response
Thank yor for youre resubmission. You have done a lot to improve youre article, but still something is missng.
-> Thank you for your detailed review. I have learned a lot and believe that the research has improved.
In the indtroduction you should explain the difference between harmony and obsessive passion more carefully. I think that to increase the number of Taekwondo players and club members and that Taekwondo becomes world-famous through this study is too ambitious.
-> As you suggested, the phrase was deleted and the realistic purpose was further described.
The contents are marked in blue. (p 2. Lines 49-54 / p 3. Lines 97-101)
Still I think there is too many hypothesis. You can remove hypothesis 3, 4 , 8 and 9.
-> This study was written to examine the structural relationship between Taekwondo players' athletic passion, sports confidence, and intention to continue exercise.
In addition, identifying the relationship between exercise passion and exercise continuity intention is a key topic of this study.
During the review, (Figure 1) was confirmed that there was an error in the research model, so I think the above revision was carried out.
Therefore, I have modified the research model, and I would like to suggest that we proceed with 10 hypotheses, reduced from 22 existing hypotheses.
Nevertheless, hypotheses 3, 4, 8, and 9 are not necessary at all, and if they must be deleted, we will review them again.
Please let us know your thoughts on this matter.
The model dos not support these hypotheis.
-> The incorrectly created Figure 1) research model has been modified. (p 5. Lines 197-205)
Table 2 : Iyt is enough to report the mean value of each variable.
-> I'm not sure what "Iyt" means, I think "Iyt" is "it".
I modified [table2].
After you have reduced the hypothesis you must cahnge the discussion as well.
-> I have reviewethe disscussiom in detail and corrected the incorrect part.
Good luck